# Enhancing Zero-shot OOD Detection with Pre-trained Multimodal Foundation Models

## Abstract

Out-of-distribution (OOD) detection is essential for reliable deployment of deep models in real-world scenarios. Advances in pre-trained multimodal foundation models have enabled **zero-shot OOD detection** using only in-distribution (ID) labels. Recent methods in this direction expand the label space with auxiliary labels to facilitate the discrimination between IDs and OODs. Inspired by the probabilistic formulation via Binomial distribution, we further discover the key factors that theoretically affect zero-shot OOD detection performance: the cardinality of the auxiliary label set, the similarity between labels and samples, and the uncertainty of the similarity scores. From the theoretical analysis, existing methods that construct fixed, single-modality auxiliary labels surely have limited effectiveness. To address these issues, we propose **Refer-OOD**, a framework that adaptively generates, filters, and retrieves multimodal references that explicitly account for these factors. It consists of three modules: reference acquisition, feature mapping, and decision module. Experiments across multiple benchmarks demonstrate that Refer-OOD consistently improves zero-shot OOD detection with both vision-language models (VLMs) and multimodal large language models (MLLMs).

## 1 Introduction

The rapid advancement of deep learning has led to significant progress in computer vision tasks such as image classification and object detection. However, despite the strong performance on in-distribution (ID) data, deep learning models still struggle with out-of-distribution (OOD) samples. Model predictions on OOD samples may be incorrect yet overconfident, undermining the reliability of these models in real-world applications [1, 2, 3]. Therefore, developing effective OOD detection methods is crucial for enhancing both model capability and safety.

Leveraging powerful feature representation and prior knowledge of pre-trained multimodal foundation models, **zero-shot OOD detection** [4] using only ID labels has garnered increasing attention. Recent methods along this direction distinguish OOD samples by expanding the label space with auxiliary OOD labels, either sampled from a semantic pool [5, 6] or generated via large language models (LLMs) [7, 8], and then classifying input images into ID/OOD groups based on CLIP [9]. Despite the great research progress, how to gather theoretically relevant auxiliary information for zero-shot OOD detection is still under-explored.

In this paper, inspired by recent works that model the OOD scores with Binomial distribution and infer the mathematical performance metric thereby [5, 6], we further discover that the performance of zero-shot OOD detection is closely related to the cardinality of label set, the similarity probabilities of ID and OOD samples within the constructed OOD label set, and the uncertainty of the similarity result. From this insight, previous methods that construct auxiliary (OOD) labels deviate from known

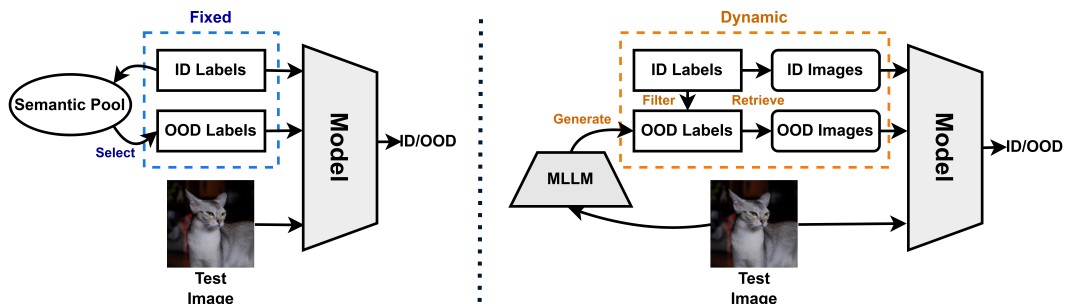

Figure 1: Comparison of two zero-shot OOD detection pipelines. Left: conventional methods mine a fixed set of OOD labels based on the ID label set from a semantic pool. Right: our proposed approach dynamically constructs reference sets by generating, filtering, and retrieving multimodal data related to the test sample at inference time.

ID labels within a single-modality framework are sub-optimal, ultimately resulting in degraded performance.

To address these issues and enhance zero-shot OOD detection, we propose **Refer-OOD**, which *adaptively* generates, filters, and retrieves *multimodal references* to increase the activation probability of OOD samples while maintaining the activation probability of ID samples. The comparison between conventional methods and Refer-OOD is illustrated in Figure 1. The entire detection process is implemented through three key modules (detailed in Figure 2): (1) a reference acquisition module for obtaining relevant references through generation, filtering and retrieval, (2) a feature mapping module that evaluates the relevance of the input image to the constructed references, and (3) a decision module that classifies samples as either ID or OOD. Theoretically, Refer-OOD can enhance OOD detection capability by dynamically integrating relevant references. Meanwhile, it is less sensitive to the reference set size.

We perform extensive experiments on coarse-grained and fine-grained OOD detection benchmarks using both traditional Vision-Language Models (VLMs) and Multimodal Large Language Models (MLLMs). The results show that Refer-OOD substantially improves model zero-shot performance in challenging OOD detection tasks. Moreover, our method consistently achieves state-of-the-art performance across various OOD detection benchmarks.

Our main contributions can be summarized as follows:

- We establish a theoretical framework for zero-shot OOD detection, based on which the key factors influencing detection performance and limitations in existing methods are identified.
- We propose Refer-OOD, a novel framework that comprehensively addresses all critical factors through adaptive label generation, similarity distribution regulation, and multimodal enhancement.
- We evaluate our method equipped with either VLMs or MLLMs and the results on both fine-grained and coarse-grained benchmarks verify the superiority of our method.

## 2 Related works

**VLMs for Traditional Out-of-Distribution Detection.** Pre-trained vision language models (VLMs) [9, 10] often require fine-tuning for effective adaptation to downstream tasks. For OOD detection, existing approaches either optimize visual or textual prompts [11, 12, 13, 14, 15, 16] or introduce OOD-specific regularization terms [14, 15, 16, 17, 18, 19]. However, these methods are computationally expensive and may undermine the generalization ability of pre-trained VLMs. Fine-tuning on ID data often leads to overfitting on seen categories, thereby reducing the model's ability to generalize to unseen ones and degrading OOD detection performance.

**Zero-shot Out-of-Distribution Detection.** Preserving VLMs' generalization ability while avoiding fine-tuning drawbacks, zero-shot OOD detection has emerged as a promising alternative. Leveraging the powerful representational capacity of pre-trained models, methods of this direction [20, 21, 22,

23, 24] bypass additional training by designing OOD detection scores to optimize the separability between IDs and OODs. Some approaches [25, 26, 27, 28] operate purely on the classification outputs of ID labels, while others [4, 6, 7, 8, 29, 30] introduce auxiliary OOD labels to recast the problem as a binary classification task distinguishing ID from OOD samples. However, constructing an appropriate OOD label set remains a non-trivial and open challenge.

**Retrieval-Augmented Generation Methods.** Retrieval-Augmented Generation (RAG) [31] combines generation with external knowledge retrieval to improve factual accuracy across various language tasks [32, 33, 34]. Recent works[35, 36, 37] extend RAG to multimodal settings by incorporating visual or auditory information, enabling richer context for generation. In this paper, we show that RAG can also enhance OOD detection by supporting retrieval-based reasoning over multimodal references.

# 3 Problem Analysis

## 3.1 Preliminaries

**Zero-shot OOD detection.** Zero-shot OOD detection aims to identify whether a test sample is in-distribution (ID) or out-of-distribution (OOD), using only ID class labels. Formally, given an ID label set $\mathcal{Y}^{\text{in}}$ of $c$ classes, and a test image from either ID or OOD domains, i.e., $x \in \mathcal{X}^{\text{in}} \cup \mathcal{X}^{\text{out}}$ with $\mathcal{X}^{\text{in}} \cap \mathcal{X}^{\text{out}} = \emptyset$, the goal is to learn a detector $h(x; \mathcal{Y}^{\text{in}}) : x \to \{\text{ID}, \text{OOD}\}$.

**OOD detection with auxiliary labels.** To facilitate the identification of OOD samples, recent works propose to augment the label space with auxiliary OOD labels, either by sampling from a semantic pool [5, 6] or generating labels via LLMs [7]. Let $\mathcal{Y}^{\text{out}} = \{y_1^{\text{out}}, \ldots, y_m^{\text{out}}\}$ denote the constructed OOD label set of size $m$. For a given image $x \in \mathcal{X}^{\text{in}} \cup \mathcal{X}^{\text{out}}$, its semantic similarity with an auxiliary label $y_i^{\text{out}} \in \mathcal{Y}^{\text{out}}$, can be computed as $s_i = \text{sim}(x, y_i^{\text{out}}) \in [0, 1]$. By applying a threshold $\psi$, this score can be converted into a binary label $b_i = \mathbb{1}_{s_i \geq \psi}$, which indicates the input is positive (OOD sample) with probability $p_i = P(s_i \geq \psi | y_i^{\text{out}}, x)$ according to the label $y_i^{\text{out}}$.

**Probabilistic approximation.** [5, 6] model the binary score $b_i$ as a random variable following Bernoulli distribution with probability $p_i$. For $x \in \mathcal{X}^{\text{in}}$, the aggregated binary score $S^{\text{in}}(x) = \sum_{i=1}^m b_i^{\text{in}}$ is then a Poisson binomial variable with probabilities $\{p_i^{\text{in}}\}_{i=1}^m$. $S^{\text{out}}$ can be defined similarly with probabilities $\{p_i^{\text{out}}\}_{i=1}^m$. According to the binomial approximation rules [38], as $m$ increases, $S^{\text{in}}$ and $S^{\text{out}}$ can be approximated as normal distributions:

$$S^{\text{in}} \sim \mathcal{N}(mp^{\text{in}}, mp^{\text{in}}(1 - p^{\text{in}}) - mv^{\text{in}}), \quad S^{\text{out}} \sim \mathcal{N}(mp^{\text{out}}, mp^{\text{out}}(1 - p^{\text{out}}) - mv^{\text{out}}), \quad (1)$$

where $p^{\text{in}} = \mathbb{E}_i[p_i^{\text{in}}], v^{\text{in}} = \text{Var}_i[p_i^{\text{in}}], p^{\text{out}} = \mathbb{E}_i[p_i^{\text{out}}], v^{\text{out}} = \text{Var}_i[p_i^{\text{out}}]$. This leads to a closed-form approximation of the false positive rate (FPR) at a target true positive rate (TPR) $\lambda \in (0, 1]$:

$$\text{FPR}_\lambda = \frac{1}{2} + \frac{1}{2} \cdot \text{erf}\left( \sqrt{\frac{p^{\text{in}}(1 - p^{\text{in}}) - v^{\text{in}}}{p^{\text{out}}(1 - p^{\text{out}}) - v^{\text{out}}}} \text{erf}^{-1}(2\lambda - 1) + \frac{\sqrt{m(p^{\text{in}} - p^{\text{out}})}}{\sqrt{2p^{\text{out}}(1 - p^{\text{out}}) - 2v^{\text{out}}}} \right), \quad (2)$$

where $\text{erf}(x) = \frac{2}{\sqrt{\pi}} \int_0^x e^{-t^2} dt$ and a lower $\text{FPR}_\lambda$ indicates better detection performance.

## 3.2 Theoretical Analysis for Performance Enhancement

*How can we minimize the FPR?*

According to Equation (2), $\text{FPR}_\lambda$ is primarily influenced by four factors: $p^{\text{in}}$, $p^{\text{out}}$, $m$, and the similarity function $\text{sim}(\cdot)$. For simplicity, we fix the factor $\lambda = 0.5$ in our analysis.

**Effect of $p_i^{\text{in}}, p_i^{\text{out}}, m$ on FPR.** Let $\zeta$ denoting the formula input to function $\text{erf}(\cdot)$ in Equation (2), the partial derivatives of $\text{FPR}_{0.5}$ with respect to $p^{\text{in}}$, $p^{\text{out}}$ and $m$ are:

$$\begin{cases} \frac{\partial \text{FPR}_{0.5}}{\partial p^{\text{in}}} = \sqrt{\frac{m}{\pi}} \cdot e^{-\zeta^2} \cdot \frac{1}{(2p^{\text{out}}(1 - p^{\text{out}}) - 2v^{\text{out}})^{\frac{1}{2}}} \geq 0, \\ \frac{\partial \text{FPR}_{0.5}}{\partial p^{\text{out}}} = -\sqrt{\frac{m}{\pi}} \cdot e^{-\zeta^2} \cdot \frac{p^{\text{out}} + p^{\text{in}} - 2p^{\text{in}}p^{\text{out}} - 2v^{\text{out}}}{(2p^{\text{out}}(1 - p^{\text{out}}) - 2v^{\text{out}})^{\frac{3}{2}}} \leq 0, \\ \frac{\partial \text{FPR}_{0.5}}{\partial m} = \frac{1}{2\sqrt{\pi m}} \cdot e^{-\zeta^2} \cdot \frac{p^{\text{in}} - p^{\text{out}}}{\sqrt{2p^{\text{out}}(1 - p^{\text{out}}) - 2v^{\text{out}}}} \leq 0, \text{when } p^{\text{in}} \leq p^{\text{out}}. \end{cases} \quad (3)$$

These results indicate that FPR$_{0.5}$ generally increases with $p^{\text{in}}$ and decreases with $p^{\text{out}}$ in most cases [5]. When $p^{\text{in}} \leq p^{\text{out}}$, increasing $m$ further reduces FPR$_{0.5}$. Therefore, an ideal $\mathcal{Y}^{\text{out}}$ should have sufficiently large size $m$, and the labels $y_i^{\text{out}} \in \mathcal{Y}^{\text{out}}$ $(i = 1, \ldots, m)$ should make ID samples yield low $p_i^{\text{in}}$ while OOD samples yield high $p_i^{\text{out}}$. Moreover, the relationship among $m, p^{\text{in}}$ and $p^{\text{out}}$ is more interdependent in practice. As $m$ increases, some $y_i^{\text{out}} \in \mathcal{Y}^{\text{out}}$ may be irrelevant to most $x \in \mathcal{X}^{\text{out}}$, causing $p^{\text{out}}$ to drop and eventually close to $p^{\text{in}}$.

Existing methods typically construct a fixed large-scale auxiliary label set $\mathcal{Y}^{\text{out}}$ by sampling or generating labels with the minimal similarity to $\mathcal{Y}^{\text{in}}$ [5, 6]. The constructed $\mathcal{Y}^{\text{out}}$ may suppress $p^{\text{in}}$ for ID inputs. However, for OOD inputs, this does not necessarily guarantee that the expectation $p^{\text{out}}$ is high enough. In practice, since $\mathcal{Y}^{\text{out}}$ is finite and fixed, there always $\exists x \in \mathcal{X}^{\text{out}}$ such that $\forall y_i^{\text{out}} \in \mathcal{Y}^{\text{out}}$, $\text{sim}(x, y_i^{\text{out}}) \ll 1$. To address the dependence between $m, p^{\text{in}}$ and $p^{\text{out}}$, [5] proposes to filter uncommon or synonymous words from $\mathcal{Y}^{\text{out}}$ to increase the possibility of $\mathcal{Y}^{\text{out}}$ being activated by $\mathcal{X}^{\text{out}}$. However, such constructing strategy still faces the inherent limitation of uncontrollable $p_i^{\text{out}}$ due to unknown $\mathcal{X}^{\text{out}}$. To address above issues, we propose to **(1) construct $\mathcal{Y}^{\text{out}}$ adaptively by generating relevant labels conditioned on the test input, rather than using a fixed set in contrast to $\mathcal{Y}^{\text{in}}$**.

**Effect of $\text{sim}(\cdot)$ on FPR.** Prior works [5, 6, 7, 25] typically compute similarity between a test image and a single reference label, which inevitably introduces high variance and limits the reliability of similarity-based decisions. We consider a more general setting where each class $y_i \in \{\mathcal{Y}^{\text{in}}, \mathcal{Y}^{\text{out}}\}$ is associated with $n_i$ *diverse references* $\{r_{i_k}\}_{k=1}^{n_i}$ (e.g., diverse text or images). The average similarity score is defined as $\bar{s}_i = \frac{1}{n_i} \sum_{k=1}^{n_i} s_{i_k} = \frac{1}{n_i} \sum_{k=1}^{n_i} \text{sim}(x, r_{i_k})$, where $\{s_{i_k}\}_{k=1}^{n_i}$ are assumed i.i.d. with mean $\mu_i^{\text{in}}$ (or $\mu_i^{\text{out}}$) and variance $\sigma_i^2$. By the central limit theorem, $\bar{s}_i^{\text{in}} \sim \mathcal{N}(\mu_i^{\text{in}}, \sigma_i^2/n_i)$, and similarly for $\bar{s}_i^{\text{out}}$. Then, the probability $p_i$ for ID or OOD inputs can be approximated by:

$$p_i^{\text{in}} = 1 - \text{erf}\left(\frac{\psi - \mu_i^{\text{in}}}{\sqrt{\sigma_i^2/n_i}}\right), \quad p_i^{\text{out}} = 1 - \text{erf}\left(\frac{\psi - \mu_i^{\text{out}}}{\sqrt{\sigma_i^2/n_i}}\right). \tag{4}$$

Reducing the variance to $\sigma_i^2/n_i$ leads to sharper similarity distributions. When $\mu_i^{\text{out}} > \psi > \mu_i^{\text{in}}$, increasing $n_i$ raises $p_i^{\text{out}}$ while suppressing $p_i^{\text{in}}$, thereby enhancing separability and reducing FPR. In contrast, existing works using a single reference increase uncertainty and yield less discriminative similarity estimates. Therefore, we propose to **(2) use multiple diverse references per class rather than relying on a single reference label**.

# 4 Method

*How can we construct a valid reference set?*

**Adaptive generation and filtering.** According to theoretical analysis (1), we propose an adaptive reference generation strategy that dynamically constructs candidate labels conditioned on test samples, controllably improving the alignment with test data distribution and increasing $p^{\text{out}}$. To stabilize $p^{\text{in}}$, a filtering mechanism is performed to discard labels overly similar to known ID classes.

**Multimodal retrieval.** According to theoretical analysis (2), to overcome the limitation of a single reference label, we introduce a modality enhancement strategy that retrieves additional image representations via an online browser API. This increases modality diversity and improves OOD detection accuracy, especially for fine-grained samples.

**Overall method.** We propose Refer-OOD, a unified OOD detection framework based on adaptive generation and modality enhancement. Our method comprises three modules (Fig. 2): (1) the Reference Acquisition Module, which obtains multimodal reference samples; (2) the Feature Mapping Module, which evaluates the relevance of $x$ to $\mathcal{Y}$; (3) the Decision Module, which determines whether the input $x$ belongs to ID or OOD.

## 4.1 Reference Acquisition Module

The reference acquisition module consists of three sequential steps: generation, filtering, and retrieval, aiming to construct high-quality textural and visual references for zero-shot OOD detection.

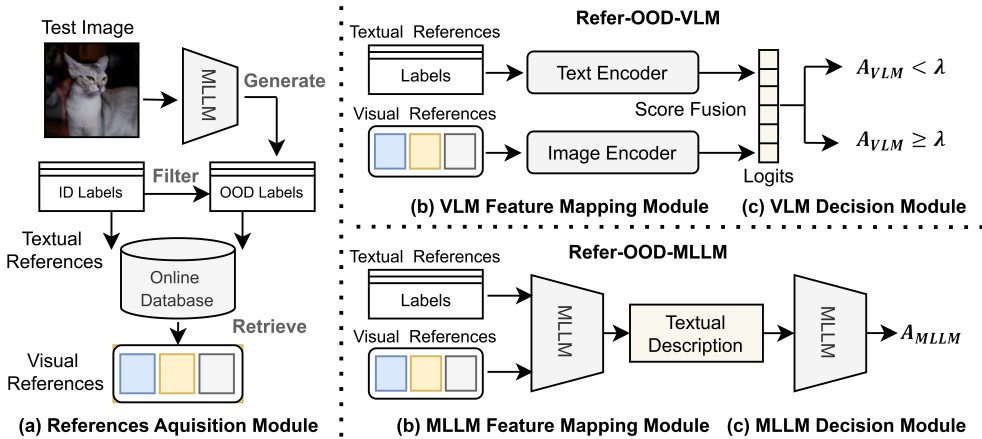

Figure 2: Detailed framework of Refer-OOD method. Refer-OOD comprises three modules: (1) Reference Acquisition Module, which obtains textual and visual references; (2) Feature Mapping Module, which evaluates the relevance of $x$ to $\mathcal{Y}$ (OOD score for VLM-based and textual description for MLLM-based method); (3) Decision Module, which determines whether the input $x$ belongs to ID or OOD.

**Generation Phase.** In the generation phase, we leverage a Multimodal Large Language Model (MLLM) $\mathcal{M}$ to produce textual labels that are semantically aligned with the input image $x$. With a prompt $\mathrm{p}_{\text{per}}$, $\mathcal{M}$ generates a set of candidate OOD labels $\widetilde{\mathcal{Y}}^{\text{out}}$:

$$\widetilde{\mathcal{Y}}^{\text{out}} = \text{TOP}_{\mathcal{Y}}(p_{\mathcal{M}}(\mathcal{Y}|x, \mathrm{p}_{\text{per}}), m), \tag{5}$$

where $p_{\mathcal{M}}(\mathcal{Y}|x, \mathrm{p}_{\text{per}})$ represents the probability distribution of MLLM generated labels $\mathcal{Y}$ given $x$ and prompt $\mathrm{p}_{\text{per}}$. Unlike prior methods that rely on fixed or predefined OOD label set, our approach generates sample-specific labels on-the-fly. This adaptive generation explicitly increases the activation probability of OOD samples on $p^{\text{out}}$, as theoretically analyzed in Section 3.2.

**Filtering Phase.** To enhance semantic distinctiveness and avoid overlap between ID and OOD labels, we apply a filtering step that removes candidate OOD labels overly similar to known ID labels $\mathcal{Y}^{\text{in}}$. This results in a refined OOD label set:

$$\mathcal{Y}^{\text{out}} = \{y_i^{\text{out}} \in \widetilde{\mathcal{Y}}^{\text{out}} \mid \max_{y_j^{\text{in}} \in \mathcal{Y}^{\text{in}}} \text{sim}(y_i^{\text{out}}, y_j^{\text{in}}) < \tau\}, \tag{6}$$

where $\tau$ denotes a predefined threshold. This filtering ensures a low similarity between ID and OOD classes, thereby stabilizing $p^{\text{in}}$.

**Retrieval Phase.** Given the final label sets $\mathcal{Y}^{\text{out}}$ and $\mathcal{Y}^{\text{in}}$, we retrieve relevant visual references from external sources such as online search engines. The retrieval process returns the top-$n_i^{\text{I}}$ images most semantically relevant to each label set:

$$\mathcal{I}^{\text{out}} = \text{TOP}_{\mathcal{I}}(p_{\text{retr}}(\mathcal{I}|\mathcal{Y}^{\text{out}}), n_i^{\text{I}}), \quad \mathcal{I}^{\text{in}} = \text{TOP}_{\mathcal{I}}(p_{\text{retr}}(\mathcal{I}|\mathcal{Y}^{\text{in}}), n_i^{\text{I}}), \tag{7}$$

where $p_{\text{retr}}(\mathcal{I} \mid \mathcal{Y})$ denotes the probability of retrieving image $\mathcal{I}$ given label set $\mathcal{Y}$.

**Optimization Design.** The reference acquisition module aims to construct reference sets that are relevant, informative and discriminative. In the generation phase, we optimize the prompt forms to guide the MLLM toward producing the highest semantically aligned labels. In the retrieval phase, we leverage the inherent ranking mechanism of the search engine to obtain top-ranked image references. This design makes the generation probability $p_{\mathcal{M}}$ and the retrieval probability $p_{\text{retr}}$ feasible and optimal.

## 4.2 Feature Mapping Module

The Feature Mapping Module aims to assess the relevance of the input sample relative to the constructed references with Vision-Language Model (VLM) or Multi-Modal Large Model (MLLM).

With similar underlying targets, these two models are different in output formats: VLM quantifies the probability differences as OOD scores, whereas MLLM generates textual descriptions that emphasize these differences.

**VLM-based Mapping Module.** We define a unified similarity function between a test sample $x$ and each (textual or visual) reference set $\{r_{i_k}\}_{k=1}^{n_i}$ corresponding to class $y_i \in \mathcal{Y}^{\text{in}} \cup \mathcal{Y}^{\text{out}}$, where $n_i = 1$ for textual reference and $n_i = n_i^{\text{I}}$ for visual references, as follows:

$$a_i(x) = \frac{|\langle I(x), \mathbb{E}_i[E(r_{i_k})]\rangle|}{|I(x)| \cdot |\mathbb{E}_i[E(r_{i_k})]|}, \tag{8}$$

where $E(\cdot)$ denotes either textual encoder $T(\cdot)$ or visual encoder $I(\cdot)$ depending on the modality of $r_{i_k}$. The OOD score for each modality follows the general form:

$$A(x) = \max_{v \in \{1,\dots,c\}} \frac{e^{a_v(x)}}{\sum_{j=1}^{m+c} e^{a_j(x)}} - \beta \max_{v \in \{c+1,\dots,m+c\}} \frac{e^{a_v(x)}}{\sum_{j=1}^{c+m} e^{a_j(x)}}, \tag{9}$$

where $\beta$ is a balancing factor. The score for comparing the test image with textual references is denoted as $A_{\text{I2T}}$, and with visual references is $A_{\text{I2I}}$. Finally, the overall multimodal detection score is fused with weight coefficient $\alpha$:

$$A_{\text{VLM}}(x) = \alpha A_{\text{I2I}}(x) + (1 - \alpha) A_{\text{I2T}}(x). \tag{10}$$

**MLLM-based Mapping Module.** The MLLM generates reasoning descriptions based on textual reference ($\mathcal{Y}^{\text{in}}$, $\mathcal{Y}^{\text{out}}$) or visual references ($\mathcal{I}^{\text{in}}$, $\mathcal{I}^{\text{out}}$), prompted by $p_{\text{rea}}$, expressed as:

$$A_{\text{MLLM}}(x) = \begin{cases} A_{\text{MLLM}}^{\text{T}}(x) = \mathcal{M}(p_{\text{rea}}||x||\mathcal{Y}^{\text{out}}||\mathcal{Y}^{\text{in}}), \\ A_{\text{MLLM}}^{\text{I}}(x) = \mathcal{M}(p_{\text{rea}}||x||\mathcal{I}^{\text{out}}||\mathcal{I}^{\text{in}}), \end{cases} \tag{11}$$

where $A_{\text{MLLM}}^{\text{T}}$ and $A_{\text{MLLM}}^{\text{I}}$ are the reasoning texts generated by comparing the input sample $x$ with the textual and visual references, respectively.

### 4.3 Decision Module

The Decision Module is responsible for determining whether a test sample $x$ belongs to ID or OOD category based on the obtained mapping results.

**VLM-based Decision Module.** For VLM model, the decision process relies on the computed detection score and a predefined threshold $\lambda$:

$$h_{\text{VLM}}(x) = \begin{cases} \text{ID}, & A_{\text{VLM}} \geq \lambda \\ \text{OOD}, & A_{\text{VLM}} < \lambda. \end{cases} \tag{12}$$

where $\lambda$ is typically set such that 95% of in-distribution (ID) data is correctly classified as ID.

**MLLM-based Decision Module.** For MLLM model, the decision process is directly based on the model-generated text $A_{\text{MLLM}}$, prompted by $p_{\text{ans}}$ for final answer:

$$h_{\text{MLLM}}(x) = \mathcal{M}(p_{\text{ans}}||x||A_{\text{MLLM}}), \tag{13}$$

which ensures that the model makes the final textual judgment (ID/OOD) based on the input sample $x$ and the relevance of corresponding references.

## 5 Experimental Analysis

### 5.1 Experimental Settings

**Datasets.** We classify OOD detection into coarse-grained and fine-grained tasks. Coarse-grained OOD detection follows the traditional setup [39], where ID and OOD belong to distinct datasets. Common ID datasets include CUB-200 [40], Stanford-Cars [41], Food-101 [42], Oxford-Pet [43], and ImageNet-1K [44], and OOD datasets include iNaturalist [45], SUN [46], Places [47], and Texture [48]. Fine-grained OOD detection is more challenging, with ID and OOD samples from the same dataset but differing at the subcategory level. Datasets are constructed by splitting categories

Table 1: Performance comparison for VLM-based methods on coarse-grained datasets.

| Method | iNaturalist | | SUN | | Places | | Texture | | Average | |
|---|---|---|---|---|---|---|---|---|---|---|
| | FPR95↓ | AUROC↑ | FPR95↓ | AUROC↑ | FPR95↓ | AUROC↑ | FPR95↓ | AUROC↑ | FPR95↓ | AUROC↑ |
| MCM | 3.27 | 99.31 | 1.68 | 99.64 | 2.63 | 99.42 | 2.91 | 99.30 | 2.63 | 99.41 |
| CLIPN | 2.20 | 99.46 | 0.88 | 99.78 | 1.83 | 99.59 | 3.11 | 99.22 | 2.00 | 99.51 |
| EOE | 0.03 | 99.99 | 0.02 | 100.0 | 0.21 | 99.94 | 0.66 | 99.76 | 0.23 | 99.92 |
| NegLabel | 0.33 | 99.91 | 0.74 | 99.78 | 1.98 | 99.46 | 1.82 | 99.51 | 1.21 | 99.66 |
| CSP | 0.25 | 99.93 | 0.28 | 99.92 | 1.67 | 99.55 | 0.98 | 99.73 | 0.79 | 99.78 |
| Refer-OOD-VLM | **0.01** | **100.0** | **0.01** | **100.0** | **0.12** | **99.97** | **0.06** | **99.99** | **0.03** | **99.99** |

Table 2: Performance comparison for VLM-based methods on fine-grained datasets.

| Method | CUB | | Stanford-Cars | | Food | | Oxford-Pet | | Average | |
|---|---|---|---|---|---|---|---|---|---|---|
| | FPR95↓ | AUROC↑ | FPR95↓ | AUROC↑ | FPR95↓ | AUROC↑ | FPR95↓ | AUROC↑ | FPR95↓ | AUROC↑ |
| MCM | 83.72 | 67.50 | 84.02 | 68.76 | 44.10 | 91.37 | 64.03 | 84.88 | 68.97 | 78.20 |
| CLIPN | 83.89 | 67.36 | 82.92 | 69.37 | 42.46 | 91.28 | 68.88 | 85.03 | 69.54 | 78.39 |
| EOE | 74.13 | 73.18 | 77.60 | 70.98 | 39.66 | **91.54** | 55.17 | 90.30 | 61.64 | 81.50 |
| NegLabel | 81.23 | 71.92 | 79.62 | 70.18 | 42.85 | 90.88 | 64.56 | 87.45 | 67.06 | 80.10 |
| CSP | 80.48 | 69.88 | 78.05 | 70.58 | 52.00 | 89.54 | 62.97 | 89.25 | 68.37 | 79.81 |
| Refer-OOD-VLM | **60.64** | **80.88** | **58.74** | **75.45** | **34.40** | 91.50 | **38.19** | **92.28** | **47.99** | **85.02** |

within CUB-200, Oxford-Pet, Food-101, and Stanford-Cars. Note that OOD categories are disjoint from ID categories.

**Experimental Setup.** We use Bing image search and a Chrome retrieval plugin for retrieval. The Qwen-vl model [49] and the CLIP [9] with ViT-B/16 serves as the MLLM and VLM backbone, respectively. More comparison results are shown in the Appendix.

**Evaluation Metrics.** For VLM-based models, we report: (1) FPR95 (false positive rate at 95% TPR) and (2) AUROC (area under the ROC curve). For MLLM-based models, we evaluate: (1) F1 score (harmonic mean of precision and recall) and (2) ACC (accuracy of ID label predictions).

**Comparative Methods.** We compare two versions of our method (named **Refer-OOD-VLM** and **Refer-OOD-MLLM** according to different feature mapping and decision modules) with state-of-the-art zero-shot OOD detection methods. For Refer-OOD-VLM, we include MCM [25], CLIPN [19], NegLabel [6], CSP [5] and EOE [7] as comparative methods. Since MLLM-based OOD detection is unexplored, we define a straightforward baseline where the model is provided with the input data and the corresponding ID labels, and asked to determine whether the sample belongs to ID.

## 5.2 Main results

**Results of VLM-based methods.** Table 1 and Table 2 present the performance of the VLM-based methods in coarse-grained and fine-grained OOD detection tasks. In the coarse-grained task, VLM-based methods achieve strong performance across all datasets, with an average AUC above 99%, benefiting from CLIP's ability to differentiate datasets with large semantic gaps. Refer-OOD-VLM outperforms all other methods on these datasets. In the fine-grained task, where ID and OOD samples are more semantically similar, all methods face increased difficulty. Nevertheless, Refer-OOD-VLM achieves more superior performance, improving FPR95 by an average of 13.65% over EOE [7].

**Results of MLLM-based methods.** Table 3 and Table 4 evaluate MLLM-based models in coarse-grained and fine-grained OOD detection tasks. "Vanilla" refers to the constructed baseline. In the coarse-grained task, both MLLM-based methods excel at distinguishing OOD samples with large semantic gaps, achieving near-perfect accuracy. Refer-OOD-MLLM further enhances performance. In the fine-grained task, Refer-OOD-MLLM outperforms the vanilla approach across most datasets and metrics, especially showing great precision gains on Stanford-Cars and Food datasets.

## 5.3 Component Analysis

**Analysis for Refer-OOD-VLM.** As shown in Table 5, both textual and visual references have a positive influence on the model's performance. The dynamic label generation strategy in Refer-OOD-VLM outperforms the fixed label approach used in the EOE method, with an AUROC increase of

Table 3: Performance comparison for MLLM-based methods on coarse-grained datasets.

| Method | iNaturalist | | SUN | | Places | | Texture | | Average | | |
|---|---|---|---|---|---|---|---|---|---|---|---|
| | Precision↑ | F1↑ | Precision↑ | F1↑ | Precision↑ | F1↑ | Precision↑ | F1↑ | Recall↑ | Precision↑ | F1↑ |
| Vanilla | 100.0 | 83.76 | 99.67 | 83.66 | 99.05 | 83.45 | 99.06 | 83.56 | 72.49 | 99.44 | 83.60 |
| Refer-OOD-MLLM | **100.0** | **90.38** | **100.0** | **90.38** | **99.24** | **90.06** | **99.32** | **90.09** | **82.75** | **99.63** | **90.16** |

Table 4: Performance comparison for MLLM-based methods on fine-grained datasets.

| Method | CUB | | | Stanford-Cars | | | Food | | | Oxford-Pet | | | Average | | |
|---|---|---|---|---|---|---|---|---|---|---|---|---|---|---|---|
| | Recall↑ | Precision↑ | F1↑ | Recall↑ | Precision↑ | F1↑ | Recall↑ | Precision↑ | F1↑ | Recall↑ | Precision↑ | F1↑ | Recall↑ | Precision↑ | F1↑ |
| Vanilla | 82.35 | 60.43 | 69.70 | 86.17 | 59.12 | 70.12 | **91.83** | 64.74 | 75.94 | 80.64 | 70.42 | 75.18 | **85.24** | 63.67 | 72.73 |
| Refer-OOD-MLLM | **85.29** | **63.50** | **72.80** | 76.59 | **79.12** | **77.83** | 79.59 | **83.87** | **81.67** | **85.48** | **92.98** | **89.07** | 81.73 | **79.86** | **80.34** |

11.14% on the CUB dataset. All modules effectively complement each other, boosting the average AUROC by 18.14% compared to the baseline.

**Analysis for Refer-OOD-MLLM.** Table 6 presents the component effectiveness analysis for Refer-OOD-MLLM, reporting detection (F1) and prediction (ACC) performance. Again, both the textual and visual references enhance model performance.

## 5.4 Case Study

**Case study for Refer-OOD-VLM.** Figures 3a to 3d show the class probability scores. For the ID sample *American bulldog*, Refer-OOD-VLM achieves high confidence on the correct label while effectively suppressing the logits of OOD labels. For the OOD sample *Beagle*, EOE misclassifies it as a similar ID category due to the absence of appropriate OOD labels. In contrast, Refer-OOD-VLM assigns high confidence to a semantically correct OOD label, reducing confusion and improving detection accuracy.

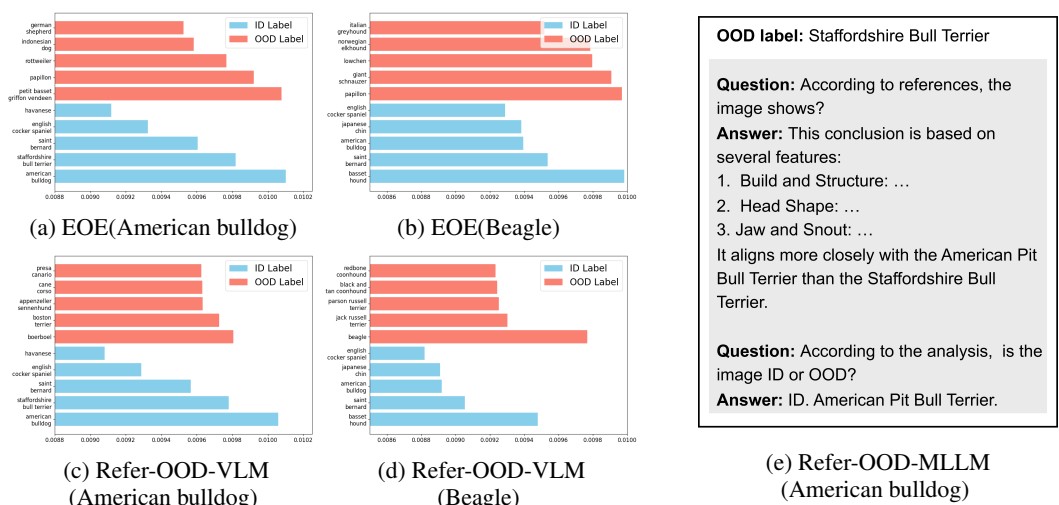

(a) EOE(American bulldog)

(b) EOE(Beagle)

(c) Refer-OOD-VLM (American bulldog)

(d) Refer-OOD-VLM (Beagle)

(e) Refer-OOD-MLLM (American bulldog)

Figure 3: Case study on Refer-OOD detection. (a)-(d) shows the class probability scores for VLM-based methods, and (e) shows Refer-OOD-MLLM result.

**Case study for Refer-OOD-MLLM.** Figure 3e illustrates one type of error made by the vanilla MLLM-based method. More illustrations in the Appendix include: (1) misclassifying ID samples as OOD, (2) predicting incorrect labels for ID samples, and (3) misclassifying OOD samples as ID. Refer-OOD-MLLM addresses these issues by incorporating valid multi-modality sets. For ID misdetection, Refer-OOD-MLLM enhances semantic understanding for accurate identification. For ID misclassification, it refines predictions using ID and OOD labels. For OOD misdetection, Refer-OOD-MLLM generates precise OOD labels to correctly classify OOD samples.

Table 5: Effectiveness of each module in Refer-OOD-VLM.

| Baseline | Fixed Textual | Dynamic Textual | Visual | CUB | | Stanford-Cars | | Food | | Oxford-Pet | |
|---|---|---|---|---|---|---|---|---|---|---|---|
| | | | | FPR95↓ | AUROC↑ | FPR95↓ | AUROC↑ | FPR95↓ | AUROC↑ | FPR95↓ | AUROC↑ |
| ✓ | | | | 80.39 | 67.34 | 63.83 | 72.07 | 55.12 | 91.79 | 83.58 | 82.93 |
| ✓ | | | ✓ | 73.53 | 68.16 | 64.89 | 76.02 | 34.35 | 92.23 | 70.15 | 84.06 |
| ✓ | ✓ | | | 67.65 | 71.88 | 62.77 | 77.24 | 50.06 | 86.94 | 47.76 | 92.27 |
| ✓ | | ✓ | | **49.02** | 83.02 | 53.19 | 75.08 | 39.44 | 92.22 | 34.33 | 92.37 |
| ✓ | ✓ | | ✓ | 65.69 | 74.41 | 60.64 | **79.62** | 49.18 | 87.66 | 41.79 | 91.91 |
| ✓ | | ✓ | ✓ | 52.94 | **85.48** | **51.06** | 77.27 | **32.83** | **92.31** | 29.85 | **93.81** |

Table 6: Effectiveness of each module in Refer-OOD-MLLM.

| Baseline | Textual | Visual | CUB | | Standford-Cars | | Food | | Oxford-Pet | |
|---|---|---|---|---|---|---|---|---|---|---|
| | | | F1↑ | ACC↑ | F1↑ | ACC↑ | F1↑ | ACC↑ | F1↑ | ACC↑ |
| ✓ | | | 69.70 | 49.01 | 70.12 | 68.08 | 75.94 | **82.65** | 75.18 | 77.41 |
| ✓ | ✓ | | 71.65 | **57.84** | 75.93 | 71.27 | **86.17** | 76.53 | 88.13 | 82.25 |
| ✓ | | ✓ | **72.80** | 55.88 | **77.83** | **75.53** | 81.67 | 70.40 | **89.07** | **85.48** |

## 5.5 Ablations

We further conduct ablation studies on the parameters, score functions, and foundation models. Please refer to the Appendix for all supporting figures and tables.

**Effect of $n_i^I$.** Figure 4 examines visual retrieval quantity $n_i^I$. Increasing $n_i^I$ improves model performance by reducing similarity variance and enhancing separability.

**Effect of $m$.** Figure 5 presents the performance of different methods as the number of generated labels increases. In contrast, Refer-OOD consistently achieves stronger results, even with fewer generated labels. Moreover, while other methods are sensitive to the value of $m$, Refer-OOD demonstrates higher robustness.

**Effect of $\beta \& \alpha$.** Figure 6 evaluates $\beta$ on fine-grained datasets. Performance improves as $\beta$ increases, with optimal results near $\beta = 1$, balancing ID and OOD labels' contributions. Figure 7 explores $\alpha$, which balances textual labels and image features in Equation (10). Introducing visual modality with well-balanced $\alpha$ outperforms single-modality approaches.

**Effect of $\tau$.** Table 11 investigates the effect of the filtering threshold $\tau$. The optimal $\tau$ typically correlates with the semantic gap between the ID and OOD datasets.

**On score functions.** Table 12 compares Refer-OOD-VLM's performance using standard scoring functions, including MSP [50], Energy [51], and MaxLogits [52]. Table 13 compares scoring function variants in Eq. 9, using max vs. sum over class logits.

**On VLMs, MLLMs and APIs.** We conduct comparative experiments across different VLMs, MLLMs and retrieval APIs. Table 14 evaluates VLM backbones including CLIP [9], ALIGN [53], and AltCLIP [54], showing that Refer-OOD consistently outperforms the comparative model across all architectures. Table 15 and Table 16 demonstrate that the results with GPT-4o are consistent with those with Qwen. Table 17 shows Refer-OOD's performance across different online retrieval APIs including Baidu and Google, which aligns with the results using Bing.

## 6 Conclusion

In this paper, we present a theoretical analysis on the zero-shot OOD detection paradigm, identifying key factors that influence detection performance, including label set size, similarity distributions, and metric uncertainty. Based on these insights, we propose Refer-OOD, a novel framework that systematically optimizes these factors through multimodal relevant references integration. Extensive experiments on both fine-grained and coarse-grained benchmarks validate the effectiveness of Refer-OOD, showing consistent improvements over prior methods.

**Broader Impacts and Limitation.** Our work promotes the reliable deployment of deep models in wide real-world scenarios, specifically on zero-shot OOD detection with pre-trained multimodal models. While our method outperforms existing approaches and maintains stable performance even with a reduced number of references, it incurs extra inference cost and possible security risks due to reference generation and retrieval.

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
