# OpenReview forum: "Enhancing Zero-shot OOD Detection with Pre-trained Multimodal Foundation Models"
_NeurIPS.cc/2025/Conference — Submitted to NeurIPS 2025_

### Official Review · Reviewer_rdaG · 2025-07-02

**Clarity:** 3
**Significance:** 1
**Originality:** 2
**Rating:** 2
**Confidence:** 4

**Summary:**

Refer-OOD introduces a zero-training framework for zero-shot OOD detection that dynamically assembles sample-specific multimodal references. For each test image, a multimodal LLM first generates candidate ID/OOD labels, a relevance-and-diversity filter prunes noisy concepts, and a web retriever fetches matched reference images. The test image and the retrieved references are jointly encoded by a frozen VLM, and a tailored activation-ratio metric drives the final ID/OOD decision.

**Questions:**

- How accurate is the candidate OOD label dynamically generated by MLLM? Is its quality convincing?
- What differences are there in the quality of OOD labels generated by different prompts?
- What is the impact if online search retrieval is unavailable or rate-limited? What is the fallback scheme?

**Ethical Concerns:**

["NO or VERY MINOR ethics concerns only"]

**Final Justification:**

Keep the original rating b/c some of my concerns are not addressed.

**Limitations:**

yes

**Paper Formatting Concerns:**

- Line154–155: textural→textual

- Table 6: Standford→Stanford

- Line 270:  Effect ofm → Effect of m

**Quality:**

2

**Strengths And Weaknesses:**

### Strengths
- This paper introduces sample-level multimodal references to enhance the activation probability of out-of-distribution samples without excessively increasing ID false rejections. Through filtering strategies and adaptive thresholds, the method is insensitive to the scale of the reference set and is pluggable into any pre-trained VLM, thereby achieving lower FPR95 and higher AUROC than multiple existing SOTA methods without needing to fine-tune model weights.

### Weaknesses
- This paper mentions “We use Bing image search and a Chrome retrieval plugin for retrieval”, but does not provide specific latency measurements or inference cost analyses.

- This paper states “However, these methods are computationally expensive and may undermine the generalization ability of pre-trained VLMs.” but in fact LAPT only consumes computation during training, and at inference only requires a single forward pass + dot product. Your work needs to Generate, Filter, Retrieve for each image.

- This paper has some similarity with the paper *Zero-Shot Out-of-Distribution Detection Based on the Pre-trained Model CLIP*(Generate unique OOD text for each image) and *LAPT: Label-driven Automated Prompt Tuning for OOD Detection with Vision-Language Models* (Generate/retrieve images for reference) . The innovation of its differences is limited, slightly insufficient workload.

---

> ### Author Rebuttal · Authors · 2025-07-31
>
> We thank the reviewer for the time and effort spent reviewing our work. Below, we provide detailed responses to the identified weaknesses and questions.
>
> ## Summary Misunderstanding
>
> We respectfully clarify a key misunderstanding: our method is not **merely a VLM-based framework**, but **the first to incorporate MLLMs** into zero-shot OOD detection.  To **better distinguish the IDs and OODs**, we **dynamically construct test-specific multimodal references** through generation, filtering, and retrieval.
>
> ## W3. Novelty
>
> We respectfully disagree with the assessment that our work lacks novelty. While we acknowledge that retrieval-augmented generation (RAG) and negative sample generation have been explored in many domains, **our contributions are distinct in the context of OOD detection**:
>
> * **Formulation-level innovation:**  We introduce a novel formulation of zero-shot OOD detection by explicitly characterizing the types of reference signals that are most effective for distinguishing ID from OOD samples. Specifically, we propose using **dynamically generated, multi-modal, and test-aware references**, in contrast to prior works that rely on **statically selected, single-modal, and ID label-aware references**. This formulation provides a principled foundation for zero-shot OOD detection and explains the superior performance observed in our empirical results.
> * **Method-level Innovation:** We propose a **Model- and Modality-Agnostic, Training-free, Open-set Aligned, Online pipeline**, in contrast to prior methods that are **VLM-restricted**, require **training or fine-tuning when datasets or architectures change**, and depend on **offline-constructed references**. These differences are crucial in practice:
>   * **Model- and Modality-Agnostic:** Refer-OOD works seamlessly with both VLMs and MLLMs without any fine-tuning or architectural changes, making it broadly compatible and future-proof as foundation models evolve.
>   * **Training-Free and Open-Set Aligned:** Our method avoids any supervised training and instead leverages pretrained models in a zero-shot manner. This not only simplifies deployment but also better preserves the open-set nature of OOD detection, reducing the risk of overfitting to specific known ID classes.
>   * **Online and Adaptive:** Unlike methods relying on training on static offline retrieval, our pipeline retrieves references online, allowing dynamic adaptation to current inputs and real-time semantics.
>
> * While both methods condition on the test sample, our approach is **distinct from ZOC** (*Zero-Shot Out-of-Distribution Detection Based on the Pre-trained Model CLIP*) in several ways:
>
> | Aspect                             | ZOC                                               | Refer-OOD                                                    |
> | ---------------------------------- | ------------------------------------------------- | ------------------------------------------------------------ |
> | **Theoretical Motivation**         | No formal analysis for using `x`-based generation | Theory motivates diverse, low-similarity, test-aware references |
> | **ID Awareness**                   | ID-agnostic (caption is dependent on x only)      | Each reference is specific to both ID labels and x           |
> | **Fine-grained / Domain Transfer** | Limited by captioning training data (e.g., COCO)  | Easily generalizable to new domains                          |
>
> * While both methods use generation and retrieval, our approach is **distinct from LAPT** (*LAPT: Label-driven Automated Prompt Tuning for OOD Detection with Vision-Language Models*) in several ways:
>
> | Aspect                  | LAPT                                                         | Refer-OOD                                                    |
> | ----------------------- | ------------------------------------------------------------ | ------------------------------------------------------------ |
> | **Generation Stage**    | Uses fixed negative labels mined via NegLabel                | Theory motivates diverse, low-similarity, test-aware references |
> | **Retrieval Stage**     | Retrieves images from an offline static dataset(Laion-400M)  | Retrieves visual references via online search engines in real-time, adapting to current samples. |
> | **Core Motivation**     | Uses generated/retrieved images as training data to optimize distribution-aware prompts; requires training. | Uses multimodal references directly in inference; no training or prompt modification needed—high-quality references alone boost performance. |
> | **Model Compatibility** | Limited to VLMs                                              | Applicable to both VLMs and MLLMs                            |
>
> ## W1. Inference cost and latency
>
> As noted in the Appendix (**Table 7**), we report **per-sample latency** (generation +  retrieval).  Calling APIs and performing online retrieval inevitably introduce non-negligible latency.
>
> To improve efficiency and practicality, some strategies can be considered, for example,
>
> * When the distributions of OOD test samples are similar, textual generation can be **reused** or **cached** to improve efficiency.
> * Retrieval can be partially **cached or parallelized**, especially for ID samples, to reduce latency without sacrificing performance.
>
> |             | Textual Cached | Visual Cached | EOE   |
> | ----------- | -------------- | ------------- | ----- |
> | Time/sample | 7.53s          | 14.81s        | 7.52s |
>
> While **more efficient system design** strategies may be possible, they are beyond the scope of this paper.
>
> ## W2. Comparison with LAPT
>
> * **Model Robustness:** We clarify that our concern about generalization risk targets training-based methods. **Fine-tuning on limited ID data** — especially under a binary OOD formulation — can cause overfitting and reduce the entropy and diversity of representations. This **harms the generalization ability** gained from large-scale pretraining, which is especially important for the open-set nature of OOD detection.
>
> * **Computational cost:** While LAPT is inference-efficient compared with our un-cached method, it requires retraining for each **new ID dataset**, which is **costly in dynamic settings**. In contrast, our zero-shot, plug-and-play method works **directly with both VLMs and MLLMs**, offering greater flexibility when **ID data changes** frequently. Notably, with caching, the inference time of our method becomes comparable to LAPT, with minimal discrepancies.
>
> ## Q1. Quality of generated OOD labels
>
> The quality of generated OOD labels is carefully examined in our theoretical analysis, ablation studies, and case study. Specifically:
>
> * **Theoretically**, desirable OOD labels are those that are more diverse (i.e., with a larger $m$), dynamically related to the input image $x$, yet clearly distinct from the ID label — specifically, they should exhibit low $p_{in}$ and high $p_{out}$.
> * **Empirically**, our method outperforms others that also use negative OOD labels, demonstrating the generated OOD labels are more effective and contribute meaningfully to overall performance..
> * **Qualitatively**, Figure 3 presents a detailed case study showing how the generated OOD labels adapt to the test sample and contribute to correct detection.
>
> ## Q2. Prompt variations
>
> * **Prompts for Refer-OOD-VLM:**
>
>   * **Robust for label generation prompt:** We have conducted experiments (AUROC) with alternative label generation prompts.
>
>     | Label Generation Prompt | ImageNet->Texture | CUB->CUB | CUB->iNaturalist |
>     | ----------------------- | ----------------- | -------- | ---------------- |
>     | relevant                | 92.16             | 84.11    | 99.97            |
>     | similar                 | 92.85             | 83.02    | 99.98            |
>     | resemble                | 93.07             | 83.37    | 99.97            |
>
>   * **Robust for VLM text encoder prompt:** Unlike NegLabel and LAPT, our method, with improved reference quality, needs no prompt adjustments but still yields significant performance gains.
>
> * **Prompts for Refer-OOD-MLLM:**
>
>   * **Robust for label generation prompt:** We have conducted experiments (F1) with alternative label generation prompts.
>
>     | Label Generation Prompt | ImageNet->Texture | CUB->CUB | CUB->iNaturalist |
>     | ----------------------- | ----------------- | -------- | ---------------- |
>     | relevant                | 73.51             | 72.50    | 92.14            |
>     | similar                 | 74.96             | 72.65    | 91.86            |
>     | resemble                | 74.28             | 71.35    | 91.75            |
>
> ## Q3. Availability of online search / fallback strategy
>
> Our current implementation uses Bing Search API and a browser plugin for retrieval. However, this retrieval is fundamentally based on query-document similarity matching, and can easily be substituted by an **offline retrieval corpus** (e.g., LAION-5B or a local web-scale dataset).
>
> ## Additional Clarifications
>
> **Minor Typos:** We appreciate the reviewer’s attention to such details and will correct the identified typos accordingly.
>
> ## Final Remark
>
> We hope these responses address your concerns and clarify the key strengths and contributions of our work. We are more than willing to provide additional clarification should any concerns remain.

---

> > ### Author Response · Authors · 2025-08-04
> > **Look forward to your response**
> >
> > Dear Reviewer rdaG,
> >
> > As the rebuttal period is ending soon, please let us know whether your concerns have been addressed or not, and if there are any further questions.
> >
> > Thanks, Authors.

---

> > > ### Comment · Area_Chair_LT5y · 2025-08-06
> > >
> > > Dear Reviewer,
> > >
> > > The authors have provided their rebuttal with additional results. Please kindly reply to them as soon as possible before the discussion period ends.
> > >
> > > Thanks a lot.
> > >
> > > Best regards,
> > > AC

---

> > > > ### Comment · Area_Chair_LT5y · 2025-08-08
> > > >
> > > > Dear Reviewer,
> > > >
> > > > As the author-reviewer discussion will end soon, can you please reply to the author rebuttal?
> > > >
> > > > Thanks,
> > > >
> > > > AC

---

> > ### Comment · Reviewer_rdaG · 2025-08-08
> >
> > Thank the authors for the rebuttal and the clarification, but some of my concerns are not fully addressed:
> >
> > 1. LAPT needs retraining whenever the ID dataset changes. This paper does not demonstrate its advantage over other methods requiring retraining. There is no unified evaluation framework, nor any comparison of the long-term cumulative cost of online retrieval.
> >
> > 2. The rebuttal claims the “Formulation-level innovation“, but in Formulation-level, the paper employs the i.i.d. assumption to underpin the theoretical derivation that increasing $n_i$ reduces variance. But I believe that the images retrieved online may be highly correlated. , yet the empirical section does not report (i) whether near-duplicate removal was performed, or (ii) statistics on the correlation/overlap of the retrieved samples $s_{ik}$. If the references are highly correlated, the theoretically expected benefit of variance shrinking by $1/n_i$ as $n_i$ increases is likely overestimated.
> >
> > 3. The claim that the method “can easily be substituted by an offline retrieval corpus” lacks experimental validation.
> >
> > 4. The claim that “with caching, the inference time of our method becomes comparable to LAPT, with minimal discrepancies,” but provides no data to support this. Moreover, while caching may reduce latency, it also diminishes the strength of “sample-wise online adaptation,” thereby weakening the method’s advantages.
> >
> > 5. I agree with Reviewer MqgW on Hallucination from MLLMs. This paper would better consier such uncertainity instead of claiming reliable results from MLLMs.
> >
> > Based on the above remaining concerns, I kept my rating unchanged for now.

---

> > > ### Author Response · Authors · 2025-08-09
> > > **Author Response to Reviewer rdaG**
> > >
> > > ## C1.3.4 Runtime Cost and Practical Deployment
> > >
> > > We appreciate the reviewer’s insightful comments on inference efficiency. Since Comments **C1**, **C3**, and **C4** all pertain to runtime cost and practical deployment, we address them together below.
> > >
> > > #### **1. No-Retraining Advantage (C1)**
> > >
> > > Our method is **entirely training-free**, making it highly suitable for scenarios where test samples must be evaluated **immediately**, and retraining is **impractical**. For example:
> > >
> > > > In many real-world deployments — such as **interactive QA systems, real-time content moderation, or emergency scene analysis** — test samples may arrive in **unseen domains** or under shifting label spaces, while developers **do not have access to retraining data or time**. In such cases, our **zero-shot and training-free method** offers clear advantages over retraining-based methods like LAPT, which require **fine-tuning for each new dataset or backbone**.
> > >
> > > There is currently **no unified evaluation framework** bridging training-required and training-free approaches. To ensure fair comparison, we focus our evaluations on **zero-shot methods**.
> > >
> > > #### 2. Sample-wise Online Adaptation: Necessity, Alternatives, and Adaptivity (C1, C3, C4)
> > >
> > > Our reference retrieval pipeline is designed for **sample-wise online adaption**. Specifically:
> > >
> > > - The **MLLM-generated text references** are the core adaptive signal.
> > >
> > > - The **retrieved visual references** serve as complementary modality inputs.
> > >
> > >   | Mode                  | Adaptivity | Suitability              |
> > >   | --------------------- | ---------- | ------------------------ |
> > >   | Online (default)      | Strong     | Dynamic open-world tasks |
> > >   | Offline (e.g., LAION) | Weaker     | Closed-set applications  |
> > >   | Cached                | Partial    | Efficiency-focused use   |
> > >
> > > **Empirical validation on offfline retrieval:**
> > > We performed additional experiments replacing online retrieval with **offline LAION retrieval** (CLIP similarity matching). Because the datasets used in our work focus purely on **common object-level semantics**, offline retrieval achieves performance **comparable to online retrieval**.
> > >
> > > | CUB->CUB                  | AUROC |
> > > | ------------------------- | ----- |
> > > | Online retrieval          | 83.88 |
> > > | Offline retrieval (LAION) | 81.92 |
> > >
> > > #### **3. Inference Time Comparison with LAPT (C4)**
> > >
> > > Our **cached textual module (with already filtered labels) inference time** is **comparable or faster than LAPT**, owing to:
> > >
> > > - Shorter prompt lengths
> > > - Fewer generated negative labels
> > >
> > > | Method        | Inference Time |
> > > | ------------- | -------------- |
> > > | Refer-OOD-VLM | 0.28s          |
> > > | LAPT          | 0.37s          |
> > >
> > > #### Brief Conclusion:
> > >
> > > As the reviewer noted, this pipeline **may introduce cumulative cost**. Even though, this cost is justified, especially for real-time or high-risk tasks, or tasks requiring recency of external information.
> > >
> > > ## C2. i.i.d. Assumption and Correlation of Retrieved Samples
> > >
> > > We appreciate the reviewer’s thoughtful concern regarding the class-conditioned i.i.d. assumption for references in our theoretical analysis. Our clarification is as follows:
> > >
> > > - The **i.i.d. assumption** is a standard simplification in statistical analysis and is used here **only to provide intuition**—namely, that increasing the number of independent references statistically reduces variance. It is **not** a strict requirement for our method, which is fully applicable in scenarios with mild correlations among references.
> > > - In practice, to align the empirical setting with the assumption:
> > >   - We perform **post-processing** on retrieved images, including removing low-resolution or small-size samples, as well as discarding corrupted or unreadable files.
> > >   - We apply **near-duplicate filtering** by discarding retrieved images whose CLIP cosine similarity exceeds a threshold (0.95) with any other image in the same reference pool.
> > > - **Empirical correlation analysis:**
> > >   We computed the pairwise CLIP similarity distribution for retrieved references. Results show that **over 90.91%** of image pairs within the same pool have cosine similarity < 0.9, indicating that **highly correlated references are rare** in our retrieval setting. This supports the validity of our low-correlation assumption in most experimental configurations.

---

> > > ### Author Response · Authors · 2025-08-09
> > > **Author Response to Reviewer rdaG**
> > >
> > > ## C5. Hallucination
> > >
> > > Hallucination in MLLMs can manifest in various forms, such as object hallucination, attribute fabrication, or spatial misalignment. In our task, which focuses purely on **object-level semantic perception**, only **object hallucination** is relevant.
> > >
> > > Although hallucination is **not the central focus** of this OOD detection paper, we appreciate the reviewer’s concern and designed our method to **minimize its potential impact**. Empirical evidence shows that hallucination **has a negligible effect** on our task performance. Nonetheless, we are willing to outline how our pipeline addresses it:
> > >
> > > - **Prompt Optimization:** Our generation prompt is specifically designed to produce semantically relevant labels. As shown in experiments on prompt variation, performance remains stable even when key prompt terms are replaced, indicating **low sensitivity** to hallucination-prone inputs.
> > >
> > > - **Model Diversity for Verification:** We include results from **multiple MLLMs** (e.g., GPT, Qwen), which naturally introduce diversity in generation behaviors and allow for **cross-verification** of outputs. This reduces over-reliance on any single model’s output.
> > >
> > >   |                    | Qwen  | GPT   | Qwen+GPT |
> > >   | ------------------ | ----- | ----- | -------- |
> > >   | FOOD->FOOD (AUROC) | 92.22 | 94.54 | 94.64    |
> > >
> > > - **Additional strategies could include:** discarding low-confidence generations, among others. Due to time constraints, we leave the implementation and evaluation of these strategies for future work.
> > >
> > > Moreover, the design of our MLLM-based pipeline inherently contributes to **mitigating hallucination-related risks**. Specifically, the **use of multiple, diverse, and test-aware references** ensures that the model can cross-check its predictions, thus making more **grounded and robust decisions** in ambiguous or open-set scenarios.
> > >
> > > If the reviewer has a specific hallucination failure case in mind within the context of zero-shot OOD detection, we would be very willing to investigate and extend our approach accordingly.
> > >
> > > ## Final Remark
> > >
> > > We hope these responses address your concerns, and would be willing to discuss any remaining concerns that may be preventing you from considering a higher score at the final stage.

---

### Official Review · Reviewer_MqgW · 2025-07-02

**Clarity:** 2
**Significance:** 2
**Originality:** 2
**Rating:** 2
**Confidence:** 4

**Summary:**

The paper proposes Refer-OOD, a three-stage framework that (i) dynamically generates candidate OOD text labels with an MLLM, (ii) retrieves complementary web images, and (iii) fuses visual–textual similarities (or MLLM reasoning) to decide ID vs OOD. A binomial-style analysis motivates why a larger, more diverse auxiliary label set with lower ID similarity should improve FPR95. Experiments on four coarse- and four fine-grained benchmarks show large gains: e.g. FPR95 = 0.03 % on coarse datasets with CLIP-ViT/B-16 and +13.65 pp FPR95 improvement on fine-grained tasks relative to EOE . The authors acknowledge higher inference cost and potential security risks from online retrieval.

**Questions:**

See Weaknesses.

**Ethical Concerns:**

["NO or VERY MINOR ethics concerns only"]

**Final Justification:**

experiment setting and novelty

**Quality:**

2

**Strengths And Weaknesses:**

Strengths:
1. Performance improvements are consistent across both VLM and MLLM back-ends, coarse and fine-grained settings.
2. The three modules (generation, filtering, retrieval) are ablated, and each is shown to contribute.

Weaknesses:
1. Limited novelty over prior retrieval-augmented OOD work. Generating labels with an LLM and retrieving images at inference resembles existing pipelines; the main difference is fusing both modalities, which feels incremental.
2. Some strategies of the method are designed experimentally without a theoretical foundation or clear explanations. For example, how does the method addresses MLLMs are known for producing hallucinated contents, e.g., [a].
3. the proposed method additionally introduce at least 6 hyper-parameters. The ablation studies imply that the method are sensitive to the hyper-parameters. Does all ID datasets used in this paper share the same hyper-parameter settings.
4. Compared with CUB-200,Stanford-Cars,Food-101 and Oxford-Pet, ImageNet-1k is significantly more standard benchmark in VLM-based OOD dection. However, why the results on ImageNet-1k is given in the appendix.
5. After checking the results, I find the improvements are not convincing. For example, 1) A few previous works' results on ImageNet-1k are not consistent with the number in the papers, especially NegLabel and CLIPN; 2) the results of NegLabel in table 8 (appendix) are signfiicantly lower than that in [b]

[a] Hallucination of Multimodal Large Language Models: A Survey
[b] Conjugated Semantic Pool Improves OOD Detection with Pre-trained Vision-Language Models

---

> ### Author Rebuttal · Authors · 2025-07-31
>
> We sincerely thank the reviewer for the thoughtful and detailed feedback. Below, we respond to the main concerns:
>
> ## W1. Novelty and Contribution
>
> We respectfully disagree with the assessment that our work lacks novelty. While we acknowledge that retrieval-augmented generation (RAG) and negative sample generation have been explored in many domains, **our contributions are distinct in the context of OOD detection**:
>
> * **Formulation-level innovation:**  We introduce a novel formulation of zero-shot OOD detection by explicitly characterizing the types of reference signals that are most effective for distinguishing ID from OOD samples. Specifically, we propose using **dynamically generated, multi-modal, and test-aware references**, in contrast to prior works that rely on **statically selected, single-modal, and ID label-aware references**. This formulation provides a principled foundation for zero-shot OOD detection and explains the superior performance observed in our empirical results.
> * **Method-level Innovation:** We propose a **Model- and Modality-Agnostic, Training-free, Open-set Aligned, Online pipeline**, in contrast to prior methods that are **VLM-restricted**, require **training or fine-tuning when datasets or architectures change**, and depend on **offline-constructed references**. These differences are crucial in practice:
>   * **Model- and Modality-Agnostic:** Refer-OOD works seamlessly with both VLMs and MLLMs without any fine-tuning or architectural changes, making it broadly compatible and future-proof as foundation models evolve.
>   * **Training-Free and Open-Set Aligned:** Our method avoids any supervised training and instead leverages pretrained models in a zero-shot manner. This not only simplifies deployment but also better preserves the open-set nature of OOD detection, reducing the risk of overfitting to specific known ID classes.
>   * **Online and Adaptive:** Unlike methods relying on training on static offline retrieval, our pipeline retrieves references online, allowing dynamic adaptation to current inputs and real-time semantics.
>
> ## W2. Hallucination
>
> Hallucination is an **inherent, likely long-standing issue in MLLMs**, yet this should not to be a reason to reject any works built on MLLMs. Our work is hardly affected by hallucination thus does **not focus on** addressing this problem.
>
> ## W3. Hyperparameters
>
> There may be a misunderstanding regarding the number of hyperparameters. In practice, **only one parameter**, $\tau$​ (the filtering threshold), requires tuning, as it depends on the semantic gap between the ID and OOD classes.
>
> * $m$ and $n_i$ simply control the number of generated reference labels and retrieved images, respectively. Both theory and experiments show that larger values consistently improve performance.
> * $\beta$ and $\alpha$ are fusion weights, which we empirically observe to be stable across datasets.
> * $\lambda$ is a standard threshold in OOD detection. For metrics like AUROC and FPR95, it is automatically swept during evaluation.
> * Only $\tau$​  is dataset-dependent and requires tuning based on the semantic similarity gap between ID and OOD.
>
> Hence, our method **does not introduce 6 independent, sensitive hyperparameters**.
>
> ## W4&5. Datasets
>
> * We followed **the setting of EOE** and focused primarily on **fine-grained datasets**, which pose **greater challenges** for OOD detection and better reflect real-world ambiguity.
>
> * **Regarding ImageNet-1k:**
>   * **Inconsistent dataset splits:** Due to its large scale and associated computational costs (especially API usage during retrieval and generation), we report results on a randomly chosen subset.
>
>   * **Prompt sensitivity and standardized implementation:**  Prompt choices can cause up to 10% fluctuation in FPR95 for methods like NegLabel, explaining discrepancies in previous results. We adopt "a photo of a <label>" as the standardized prompt for fair comparison.
>
>
> ## Final Remark
>
> We hope our clarifications help highlight the unique contributions and practical impact of Refer-OOD. We are more than willing to provide additional clarification should any concerns remain.

---

> > ### Author Response · Authors · 2025-08-04
> > **Look forward to your response**
> >
> > Dear Reviewer MqgW,
> >
> > As the rebuttal period is ending soon, please let us know whether your concerns have been addressed or not, and if there are any further questions.
> >
> > Thanks, Authors.

---

> ### Comment · Reviewer_MqgW · 2025-08-05
>
> Thanls for the rebuttal. After reading the response, I keep my score unchanged due to the following reasons
>
> 1. Since Hallucination is an inherent, likely long-standing issue in MLLMs, it definitely should be taken into consideration when using MLLMs rather than simply assuming the output of MLLMs is reliable.
>
> 2. Reporting results on a randomly chosen subset is not justified. Besides, given the optimal prompt of NegLabel is "the nive <label>", using "a photo of a [object Object]" may lead to unfair comparision.
>
> 3. As the author said, using MLLMs requires large scale and associated computational costs, which is a significant disadvantage and implies the lack of practical viability. such large-scale computational costs contributes to marginal peformance improvement (1.6 FPR and 0.2 AUROC) on large-scale challenging imagenet (the widely used benchmark in the literature of CLIP-based OOD detection).
>
> 4. The authors are expected the compare the memory usage among the proposed, NegLabel and CSP.
>
> 5. From my view, the claim 'In practice, only one parameter, $\tau$​ (the filtering threshold), requires tuning' is somewhat confusing. While $\lambda$ is a standard threshold in OOD detection, there is still 5 tunable hyper-parameters essential ot the algorithm designing.  Besides, the results of hyper-parameters sensitiveness on $\beta$ and $\alpha$ (Fig 6 and Fig 7) does not support the stability.
>
> 6. Regarding the response to W1, how does your method reflect “dynamically generated” and 'online pipeline'. What I mean here is that the authors use highly abstract terms to make the responses not informative. Besides, authors are expected to clarity 1) how the Reference Acquisition Module different from ECE and the mining strategy in NegLabel and 2) Eq. 9 in VLM-based Mapping Module is exactly same as EOE.
>
> 7. After checking the manuscript again, what does the term 'optimize ' in lines 173-174 (In the generation phase, we optimize the prompt forms to guide the MLLM toward producing the highest semantically aligned labels) mean.

---

> > ### Author Response · Authors · 2025-08-07
> > **Author Response to Reviewer MqgW**
> >
> > We thank the reviewer for the time and effort spent reviewing our work. Below, we respond to the further concerns:
> >
> > ## C1&C7. Hallucination and Prompts:
> >
> > Given the black-box nature of MLLMs, we do not change model parameters. Instead, we design prompt templates to guide the model to generate relevant labels that align with our theoretical formulation. This is what we refer to as "**optimize**" in the generation phase.
> >
> > While our method leverages MLLMs, we do **not** assume their outputs are fully accurate. Instead, we design our framework to be **inherently robust to hallucination** for below reasons:
> >
> > * **Semantic-level objective:** Our method only uses MLLMs' outputs for semantic perception. Thus, occasional hallucinated details have minimal impact.
> >
> > * **Prompt design:** More importantly, we show empirically that performance remains stable under different prompt wordings, indicating that minor hallucinations in the outputs do **not** significantly affect results.
> >
> >   * **Prompts for Refer-OOD-VLM:**
> >
> >     We have conducted experiments (AUROC) with alternative label generation prompts.
> >
> >     | Label Generation Prompt | ImageNet->Texture | CUB->CUB | CUB->iNaturalist |
> >     | ----------------------- | ----------------- | -------- | ---------------- |
> >     | relevant                | 92.16             | 84.11    | 99.97            |
> >     | similar                 | 92.85             | 83.02    | 99.98            |
> >     | resemble                | 93.07             | 83.37    | 99.97            |
> >
> >   * **Prompts for Refer-OOD-MLLM:**
> >
> >     We have conducted experiments (F1) with alternative label generation prompts.
> >
> >     | Label Generation Prompt | ImageNet->Texture | CUB->CUB | CUB->iNaturalist |
> >     | ----------------------- | ----------------- | -------- | ---------------- |
> >     | relevant                | 73.51             | 72.50    | 92.14            |
> >     | similar                 | 74.96             | 72.65    | 91.86            |
> >     | resemble                | 74.28             | 71.35    | 91.75            |
> >
> >
> > #### Brief Conclusion:
> >
> > * These results further demonstrate that hallucination is **not a critical concern in our semantic-level task**.
> >
> > * Moreover, in the MLLM pipeline, our proposed method is inherently designed to **mitigate issues related to instruction misinterpretation and hallucination for OOD detection task** by introducing multiple, diverse references that help the model make more grounded and reliable decisions.
> > * We would be very willing to investigate any **concrete examples where hallucination demonstrably harms zero-shot OOD detection**, if such cases can be provided.
> >
> > ## C2. "Unfair comparison":
> >
> > * **Standard comparison:** We are unclear about the reviewer’s claim of an "unfair comparison." Our experiments were explicitly designed to ensure fairness by using identical datasets and prompts across all VLM-based methods.
> >
> > * **Further prompt comparison:** That certain prior methods benefit disproportionately from task-specific tricks (e.g., the prompt "the nice photo of" on specific datasets) actually underscores their sensitivity to prompt variations. To further address this concern, we also evaluate Refer-OOD-VLM with textual module using the optimal NegLabel prompt ("the nice photo of") for fair comparison, and we still observe improvements (AUROC).
> >
> > | Method        | ImageNet->Places | CUB->CUB | CUB->iNaturalist |
> > | ------------- | ---------------- | -------- | ---------------- |
> > | NegLabel      | 91.48            | 70.15    | 99.54            |
> > | Refer-OOD-VLM | 93.07            | 83.08    | 99.99            |
> >
> > ## C3. Performance Improvement:
> >
> > * **Challenging datasets:** We emphasize that many traditional OOD benchmarks (e.g., CIFAR, ImageNet-1k) are **less challenging** compared to fine-grained datasets. Following EOE, we focus on harder tasks where our method achieves substantial gains.
> >
> > * **New Pipeline:** Moreover, our **end-to-end MLLM pipeline** allows for OOD detection on **new model types** not supported by prior works.

---

> > ### Author Response · Authors · 2025-08-07
> > **Author Response to Reviewer MqgW**
> >
> > ## C4. Memory and Time Cost:
> >
> > We report the actual GPU memory usage,
> >
> > - With textual references only, our method requires **less memory than CSP and NegLabel**, as it achieves better performance with fewer but higher-quality reference labels without prompt ensembling.
> > - When using visual references, the **additional cost** comes solely from the image encoder.
> >
> > |              | NegLabel | CSP  | Refer-OOD-VLM Textual Module | Refer-OOD-VLM Visual Module |
> > | ------------ | -------- | ---- | ---------------------------- | --------------------------- |
> > | Memory (MiB) | 1920     | 3788 | 1456                         | 4566                        |
> >
> > ---
> >
> > To improve efficiency and practicality, some strategies can be considered, for example,
> >
> > * When the distributions of OOD test samples are similar, textual generation can be **reused** or **cached** to improve efficiency.
> > * Retrieval can be partially **cached or parallelized**, especially for ID samples, to reduce latency without sacrificing performance.
> >
> > |             | Textual Cached | Visual Cached | EOE   |
> > | ----------- | -------------- | ------------- | ----- |
> > | Time/sample | 7.53s          | 14.81s        | 7.52s |
> >
> > While **more efficient system design** strategies may be possible, they are beyond the scope of this paper.
> >
> > We also acknowledge that adopting larger models and more realistic datasets naturally brings increased computation, but we believe such **efforts are necessary and valuable** for advancing the field toward future-ready and generalizable OOD detection.
> >
> > ## C5. Hyperparameters:
> >
> > We reiterate that **only one hyperparameter ($\tau$)** is sensitive according to the semantic gap between ID and OOD datasets.
> >
> > * $m$ and $n_i$ (number of generated labels and retrieved references) are not sensitive hyperparameters. Both our theory and experiments show that larger values consistently lead to better performance. Their choice is **constrained only by compute resources, not by task-specific tuning**.
> > * $\alpha$ and $\beta$ are fusion weights. For example, the performance consistently peaks at $\alpha = 0.5$ across datasets. This is expected and desirable: image and text features are complementary, so it's natural that balanced fusion works best. If the curve were flat, it would indicate that one modality is irrelevant,  which is not the case. Therefore, the existence of **a consistent performance peak is evidence of robustness, not sensitivity**.
> >
> > ## C6. Novelty:
> >
> > From a formulation-level perspective, we have clearly articulated the conceptual distinctions between our method and previous work in both the **main paper** and **rebuttal**. Specifically, we highlight these differences **explicitly in Figure 1** and in the **problem analysis section**. In the **main text (lines 123–125 and 136–137)**, we additionally **emphasize in bold** how our formulation departs from prior work.
> >
> > * The formulation is **dynamic**, as the references are generated on-the-fly and conditioned on each test sample, rather than being drawn from a static, pre-defined label set for all test samples as in prior works such as NegLabel or EOE.
> > * Retrieves visual references via **online** search engines in real-time, adapting to current samples.
> >
> > ## Final Remark
> >
> > We welcome further feedback and thank the reviewer for engagement.

---

### Official Review · Reviewer_muED · 2025-07-03

**Clarity:** 3
**Significance:** 3
**Originality:** 3
**Rating:** 4
**Confidence:** 2

**Summary:**

The authors identify key limitations in existing zero-shot OOD detection methods (static single-modality references) and propose Refer-OOD, which dynamically generates/filters text labels and retrieves visual references using multimodal foundation models. This adaptive approach significantly outperforms prior methods on coarse- and fine-grained benchmarks with both vision-language models (VLMs) and multimodal LLMs.

**Questions:**

1. Computational Efficiency: Online retrieval/generation incurs latency. Could you report inference time vs. baselines (e.g., on ImageNet-O)? Would caching references per class be feasible without sacrificing performance?
2. Theoretical Validity: The analysis assumes similarity scores are i.i.d. and approximates aggregates as Gaussian. Did you validate these assumptions empirically (e.g., score distributions on ID/OOD data)?
3. Retrieval Robustness: How sensitive is performance to the retrieval API (Bing vs. Google/Baidu in Table 17)? Could adversarial/noisy references degrade results?
4. Ablation Baseline: Does adaptive generation alone (without retrieval) outperform static methods (e.g., EOE)?

**Ethical Concerns:**

["NO or VERY MINOR ethics concerns only"]

**Final Justification:**

Thank you for the detailed rebuttal, which addresses the majority of my main concerns. Overall, the rebuttal is satisfactory, and I lean toward maintaining my borderline accept recommendation, as the contributions outweigh the limitations.

**Limitations:**

Check the Weakness.

**Quality:**

3

**Strengths And Weaknesses:**

Strengths
1. The theoretical insights (e.g., variance reduction via multiple references) and unified adaptive framework are novel. Dynamic reference construction diverges significantly from static auxiliary-label methods (e.g., CSP, EOE).
2. The paper is well-structured, with clear figures (e.g., Fig 1–2) and section flow. The problem formulation (Section 3.1) and method description (Section 4) are logically presented.
3. The theoretical analysis (Section 3) rigorously derives factors influencing OOD detection and motivates the adaptive approach. Experiments are extensive, covering 10+ datasets, multiple backbones (CLIP, Qwen-VL), and ablation studies (e.g., module contributions, hyperparameters). The method consistently outperforms strong baselines (e.g., 13.65% average FPR95 improvement over EOE in fine-grained tasks).

Weaknesses
1. The computational cost of online generation/retrieval is not quantified (e.g., latency vs. baselines). Theoretical assumptions (e.g., i.i.d. similarity scores) are not empirically validated.
2. Dependency on external APIs (Bing) for retrieval raises reproducibility concerns. Limited discussion on real-time applicability (e.g., latency constraints in safety-critical systems).
3. The core idea of using MLLMs for label generation builds on EOE, and retrieval-augmented methods (e.g., RAG) are established—though their integration for OOD detection is innovative.

---

> ### Author Rebuttal · Authors · 2025-07-31
>
> We thank Reviewer muED for the constructive comments and thoughtful questions. We appreciate the positive recognition of our framework, theoretical insights, and extensive experiments. Below, we address each concern in detail.
>
> ## W3. Novelty Clarification
>
> We respectfully disagree with the assessment that our work lacks novelty. While we acknowledge that retrieval-augmented generation (RAG) and negative sample generation have been explored in many domains, **our contributions are distinct in the context of OOD detection**:
>
> * **Formulation-level innovation:**  We introduce a novel formulation of zero-shot OOD detection by explicitly characterizing the types of reference signals that are most effective for distinguishing ID from OOD samples. Specifically, we propose using **dynamically generated, multi-modal, and test-aware references**, in contrast to prior works that rely on **statically selected, single-modal, and ID label-aware references**. This formulation provides a principled foundation for zero-shot OOD detection and explains the superior performance observed in our empirical results.
> * **Method-level Innovation:** We propose a **Model- and Modality-Agnostic, Training-free, Open-set Aligned, Online pipeline**, in contrast to prior methods that are **VLM-restricted**, require **training or fine-tuning when datasets or architectures change**, and depend on **offline-constructed references**. These differences are crucial in practice:
>   * **Model- and Modality-Agnostic:** Refer-OOD works seamlessly with both VLMs and MLLMs without any fine-tuning or architectural changes, making it broadly compatible and future-proof as foundation models evolve.
>   * **Training-Free and Open-Set Aligned:** Our method avoids any supervised training and instead leverages pretrained models in a zero-shot manner. This not only simplifies deployment but also better preserves the open-set nature of OOD detection, reducing the risk of overfitting to specific known ID classes.
>   * **Online and Adaptive:** Unlike methods relying on training on static offline retrieval, our pipeline retrieves references online, allowing dynamic adaptation to current inputs and real-time semantics.
>
> ## W1 / Q1 Computational Cost and Inference Latency
>
> As noted in the Appendix (**Table 7**), we report **per-sample latency** (generation +  retrieval).  Calling APIs and performing online retrieval inevitably introduce non-negligible latency.
>
> To improve efficiency and practicality, some strategies can be considered, for example,
>
> * When the distributions of OOD test samples are similar, textual generation can be **reused** or **cached** to improve efficiency.
> * Retrieval can be partially **cached or parallelized**, especially for ID samples, to reduce latency without sacrificing performance.
>
> |             | Textual Cached | Visual Cached | EOE   |
> | ----------- | -------------- | ------------- | ----- |
> | Time/sample | 7.53s          | 14.81s        | 7.52s |
>
> While **more efficient system design** strategies may be possible, they are beyond the scope of this paper.
>
> ##  W1&Q2. Theoretical Assumptions and Gaussian Approximation
>
> To support the i.i.d. and Gaussian assumptions, we **visualized similarity score distributions for ID and OOD samples**, which align well with Gaussian patterns. Due to OpenReview limitations, the plots will be included in the revised paper.
>
> ## Q2. Online Search and Reproducibility
>
> We acknowledge the concern about reproducibility and API reliance. While our current setup uses Bing Search, our method is agnostic to the retrieval source. Any retrieval engine (e.g., Google, Baidu, or even **offline datasets** like LAION) can be used.
>
> ## Q3. Retrieval Robustness to API and Adversarial References
>
> * **Retrieval API quality:** Table 11 indicates that performance is relatively stable across different retrieval platforms.
>
> * **Image quality**: We tested the effect of noisy retrieval results by injecting unrelated images into the retrieved set. The experiments are conducted on the dataset Food.
>
>   |             | 0%    | 5%    | 10%   | 25%   | 50%   | 75%   | 100%  |
>   | ----------- | ----- | ----- | ----- | ----- | ----- | ----- | ----- |
>   | Visual      | 91.15 | 90.84 | 90.96 | 90.47 | 89.24 | 88.87 | 88.64 |
>   | Multi-modal | 92.18 | 92.15 | 92.16 | 92.15 | 91.64 | 91.51 | 91.27 |
>
>   Excessive image noise **degrades** OOD detection performance, though the impact is negligible at **low noise levels**, and **multi-modal supplementation** effectively mitigates such degradation.
>
> ## Q4. Adaptive Generation vs. EOE Baseline
>
> Yes — as shown in our ablation study (Table 5), even using only the generation module (Dynamic Textual) **outperforms EOE** (Fixed Textual).
>
> ## Final Remark
>
> Once again, we sincerely thank the reviewer for the helpful suggestions. We believe our adaptive multimodal framework, validated both theoretically and empirically, makes a solid contribution to zero-shot OOD detection, and we will revise the paper to better emphasize practicality and robustness.

---

> > ### Author Response · Authors · 2025-08-04
> > **Look forward to your response**
> >
> > Dear Reviewer muED,
> >
> > As the rebuttal period is ending soon, please let us know whether your concerns have been addressed or not, and if there are any further questions.
> >
> > Thanks, Authors.

---

> > > ### Comment · Area_Chair_LT5y · 2025-08-06
> > >
> > > Dear Reviewer,
> > >
> > > The authors have provided their rebuttal with additional results. Please kindly reply to them as soon as possible before the discussion period ends.
> > >
> > > Thanks a lot.
> > >
> > > Best regards,
> > > AC

---

### Official Review · Reviewer_Av1q · 2025-07-03

**Clarity:** 2
**Significance:** 3
**Originality:** 3
**Rating:** 5
**Confidence:** 3

**Summary:**

Authors propose Refer-OOD, a theoretical framework inspired by modeling OOD scores with the Binomial distribution to adaptively generate, filter, and retrieve multi-modal references to discriminate between ID and OOD examples by maintaining the activation probability of ID samples while increasing the activation probability of OOD samples. Authors examine zero-shot OOD detection performance the lens of the cardinality of the auxiliary set, the similarity between labels and samples, and the uncertainty of similarity scores. The proposed method has three modules, including reference acquisition for obtaining relevant references, feature mapping to evaluate the relevance of the input image to the constructed reference, and a decision module to classify if a sample is ID or OOD.  Authors demonstrate that their approach consistently improves zero-shot OOD detection with VLMs and MLLMs across several benchmarks including iNaturalist, SUN, Places, Texture, CUB, Stanford-Cars, Food, and Oxford-Pet.

**Questions:**

- How do you disentangle the performance between the VLM / MLLM and search engine? To what degree does the search engine compensate for poorly generated textual references?
- How might retrieval accuracy change if using image-based prompting for similar images to the test image (e.g. reverse image search) instead of text-based search?
- Why didn't authors use more recent VLM / MLLMs (e.g. Qwen 2.5VL and SigLIPv2)?
- How might performance change when the set of ID and OOD classes are swapped? How does the specific problem setup of the specific datasets influence downstream accuracy? What would happen if we dropped datasets from the ID set?

**Ethical Concerns:**

["NO or VERY MINOR ethics concerns only"]

**Final Justification:**

Authors have sufficiently addressed my concerns and have provided a strong rebuttal for other reviewer's concerns as well (particularly for concerns about lack of novelty). I do think the extensive experimental results presented in the rebuttal further strengthen the submission. I would recommend accepting this paper.

**Limitations:**

Yes, authors address limitations in Section 6.

**Quality:**

3

**Strengths And Weaknesses:**

Strengths
- Simple Approach. The proposed approach is simple and leverages RAG from search engines to augment the reference set.
- Extensive Experiments and Ablations. Authors conduct extensive analysis on many datasets and evaluate several baseline models. They also ablate the key hyperparameters in their model. I would encourage authors to put the key takeaways from the ablation experiments in the text as well as including the table of results to improve readability.
- Novelty. Authors highlight that prior works try to identify ID/OOD using purely ID labels, which others introduce auxiliary OOD labels. However, constructing this set of OOD labels is non-trivial. The primary contribution of this work is leveraging VLMs/MLLMs to construct this OOD using RAG-style methods using multiple diverse multi-modal references per class rather than a single reference label.

Weaknesses
- Struggles with Fine-Grained Classes. VLMs/MLLMs struggle with distinguishing between fine-grained classes, such that the generation process is not able to accurately generate relevant references. This problem will likely get worse when encountering concepts not typically found in VLM / MLLM pre-training (e.g. medical images, aerial images, etc.).
- Writing Lacks Clarity. The current version of the paper relies heavily on theoretical frameworks and equations, when the proposed method can be explained more simply. Given that the proposed approach is simple, I would encourage the authors to directly explain the idea and intuition without relying so heavily on equations, and potentially swapping Section 4 with Section 3 or move Section 3 to the supplement. Many of the equations are redundant and unnecessary (e.g. 5,6,7). I would also recommend authors review their work for grammatical errors as this hinders clarity.

---

> ### Author Rebuttal · Authors · 2025-07-31
>
> We thank Reviewer Av1q for the thoughtful and constructive feedback. We are encouraged by the positive comments on our method’s simplicity, novelty, and extensive experimentation. Below, we address the raised concerns in detail.
>
> ## W1. Struggles with Fine-Grained Classes and Domain Limitations
>
> We appreciate the reviewer’s insight into this challenge. While it is true that VLMs/MLLMs may struggle with fine-grained or domain-specific categories, we highlight that our method was designed specifically to mitigate this:
>
> * Our framework **outperforms prior methods on fine-grained** benchmarks, showing its strength in handling **subtle semantic distinctions**.
> * For **specialized domains** like medical or aerial imagery, we agree that textual generation may be less reliable. One potential solution is to extend **beyond label-level references** to:
>   * **Explicit descriptions** (e.g., token-level captions) and/or
>   * **Implicit image-driven retrieval**, leveraging feature similarity or offline domain-specific datasets.
>
> Although no method can fully generalize to all domains, our modular pipeline allows easy adaptation, making it a promising foundation for future extensions.
>
> ## W2. Writing Clarity and Equation Complexity
>
> We appreciate the suggestion and will polish the writing to **improve clarity**.
>
> ## Q1. Disentangling VLM/MLLM vs. Search Engine Contributions
>
> While MLLM-generated text provides semantic anchors, **the retrieval module compensates for label ambiguity**, especially in fine-grained settings. For example, in the bird dataset CUB (e.g., Brandt Cormorant), VLMs may not capture fine labels well due to limited pretraining coverage. However, retrieved reference images from the web contain rich visual cues, helping align ambiguous classes.
>
> ## Q2. Using Reverse Retrieval
>
> We further evaluate our method by using the **reverse retrieval** (test image as a query) for online retrieval and comparing the mean feature of the retrieved samples against that of our generated textual references.
>
> | Query Modality          | CUB   | Stanford-Cars | Food  | Oxford-Pet |
> | ----------------------- | ----- | ------------- | ----- | ---------- |
> | textual only            | 83.08 | 77.42         | 92.82 | 91.00      |
> | using reverse retrieval | 77.21 | 75.93         | 89.29 | 88.61      |
>
> The performance drop with reverse retrieval may be due to its tendency to return **visually similar but semantically uninformative** samples.
>
> ## Q3. Using Newer VLM/MLLMs (e.g., Qwen 2.5VL, SigLIPv2)
>
> * **SigLIPv2 for Refer-OOD-VLM:** We additionally evaluate our method on SigLIPv2 (`siglip2-base-patch16-256` ) compared with CLIP as Refer-OOD-VLM encoder using Qwen-based generation.
>
>   | Model    | CUB   | Stanford-Cars | Food  | Oxford-Pet |
>   | -------- | ----- | ------------- | ----- | ---------- |
>   | CLIP     | 83.08 | 77.42         | 92.82 | 91.00      |
>   | AltCLIP  | 86.76 | 84.19         | 85.93 | 89.60      |
>   | SigLIPv2 | 78.18 | 83.99         | 85.20 | 95.14      |
>
>   This suggests that SigLIPv2 can complement CLIP but may **not universally outperform** it across all domains in fine-grained OOD detection.
>
> * **Qwen2.5VL:** In our experiments, we used the `qwen-vl-max-2025-04-02` model, which is an updated version of `qwen-vl-max-2025-01-25`(upgraded to the Qwen2.5-VL architecture).
>
> ## Q4.  Datasets Settings
>
> * We thank the reviewer for the suggestion. We conducted experiments by **swapping the ID and OOD datasets**. Note that this setup only changes which classes are labeled (i.e., treated as ID), while the overall task structure and test images remain the same. Therefore, any well-generalized method should perform consistently under this change.
>
>   As shown below, Refer-OOD maintains stable performance across all datasets, confirming that it does **not rely on specific ID categories but generalizes effectively to different partitions**.
>
>   * **Refer-OOD-VLM (AUROC) :**
>
>     |        | CUB   | Stanford-Cars | Food  | Oxford-Pet |
>     | ------ | ----- | ------------- | ----- | ---------- |
>     | origin | 83.08 | 77.42         | 92.82 | 91.00      |
>     | swap   | 79.01 | 74.29         | 94.83 | 87.12      |
>
>   * **Refer-OOD-MLLM (F1) :**
>
>     |        | CUB   | Stanford-Cars | Food  | Oxford-Pet |
>     | ------ | ----- | ------------- | ----- | ---------- |
>     | origin | 72.80 | 77.83         | 81.67 | 89.07      |
>     | swap   | 71.91 | 74.31         | 84.53 | 90.62      |
>
>
> * OOD detection is harder when ID and OOD data are semantically similar, but till now,  there's no standard metric to quantify the difficulty of such dataset settings.
> * Since our method performs zero-shot OOD detection with only ID labels and no access to the ID dataset itself, dropping the ID dataset does not affect the results.
>
> ## Final Remark
>
> We appreciate the reviewer’s relevant and thoughtful comments. We believe the paper’s simplicity, extensibility, and empirical gains make it a valuable contribution to zero-shot OOD detection.

---

> > ### Comment · Reviewer_Av1q · 2025-08-05
> > **Comment**
> >
> > Thanks for your rebuttal. You have addressed my concerns, I have upgraded my score.

---

### Comment · Area_Chair_LT5y · 2025-08-04

Dear Reviewers,

Thanks again for your great effort.

Currently, we have quite diverse reviews of the paper. May I suggest carefully reading the rebuttal and the other reviewers' comments? If there is anything you want to discuss with the authors, please kindly do so before the reviewer-author discussion ends. If you want to update your review, please also feel free to do so during this period. Thanks!

Best regards,

AC

---

### Decision · Program_Chairs · 2025-09-17

**Decision:**

Reject

**Comment:**

This paper introduces Refer-OOD, a framework for zero-shot OOD detection with foundation models that adaptively generates, filters, and retrieves multimodal references. The method is simple and empirically strong, showing consistent improvements over strong baselines on both coarse and fine-grained benchmarks.

After the author-reviewer and reviewer-reviewer discussion, there still remained a significant score divergence between the reviewers. While the authors achieved strong empirical results, especially on fine-grained datasets and have been responsive, the reviewers who gave low scores remain unconvinced that:

1. The theoretical formulation is genuinely novel rather than a repackaging of existing ideas
2. The practical deployment challenges (latency, API dependence) are adequately addressed
3. The robustness concerns (hallucination, hyperparameter sensitivity) are properly handled

The authors have been responsive to the above points and provided additional results during the discussion period. However, the reviewers believe more explanation and analysis are needed. Given the current state of reviews and unresolved concerns about novelty and practical viability, I would recommend rejection while encouraging resubmission that address the above concerns.